# Application of Whey Protein-Based Emulsion Coating Treatment in Fresh-Cut Apple Preservation

**DOI:** 10.3390/foods12061140

**Published:** 2023-03-08

**Authors:** Ying Xin, Chenhao Yang, Jiahao Zhang, Lei Xiong

**Affiliations:** 1College of Food Science and Technology, Henan University of Technology, Zhengzhou 450001, China; soojinelk@gmail.com; 2Culinary and Food College, Zhengzhou Tourism College, Zhengzhou 450009, China; jiahaocham@163.com; 3Quality Control Centre, Henan Qinre Holding Group Limited Company, Xinxiang 453000, China

**Keywords:** edible film, water resistant, whey protein-based emulsion coating, fresh-cut apple, storage

## Abstract

Fresh-cut fruit requires an edible and water-resistant coating to remain fresh. This article investigated the effects of transglutaminase (TGase) and sunflower oil on the water-resistant characteristics, mechanical properties, and microstructure of a whey protein-based film. The whey protein-based emulsion coating’s preservation effect on fresh-cut apples was confirmed. According to the findings, sunflower oil (added at 1.5% *w*/*w*) could interact with β-lactoglobulin, α-lactoglobulin dimer, and β-lactoglobulin dimer to form emulsion droplets that are evenly dispersed throughout the protein film. This effect, combined with the covalent cross-linking of TGase, significantly improves the films’ microstructure, mechanical properties, and water resistance. However, too much and unevenly distributed sunflower oil (add 3% *w*/*w*) partially prevented the covalent cross-linking of TGase, reducing the elongation at the break of the composite film. In the fresh-cut apple storage experiment, the whey protein-based emulsion coating treatment significantly reduced the weight loss rate and browning index of fresh-cut apples by 26.55% and 46.39%, respectively. This was accomplished by the coating treatment significantly inhibiting the respiration rate increase, PPO and CAT activity enhancement, H_2_O_2_ production, and MDA accumulation. This research provides practical, technical, and theoretical guidance for the preservation of fresh-cut fruit.

## 1. Introduction

Fresh-cut fruits and vegetables are trimmed, peeled, and cut into fully usable products. Following further packaging, consumers can receive fresh items that are convenient, nutritious, and tasty [1]. Fresh-cut fruits and vegetables, in addition to being clean, hygienic, and ready to eat, have less pesticide residue, which can increasingly meet people’s higher expectations for foods. However, mechanical damage to fresh-cut fruits and vegetables during processing causes fluid loss and the breakdown of the natural tissue structure, reducing their resistance to the environment [2]. When intact fruits and vegetables are destroyed, the exposed and oxidized tissue cells are more exposed to oxygen, which accelerates respiration and causes nutrients to be lost, browning, and microbial infection [3]. Therefore, extending the shelf life of fresh-cut fruits and vegetables while maintaining the quality of the product has become a hot topic in contemporary research.

An edible coating can form a selective permeability barrier on the surface of fruits and vegetables, inhibit transpiration and respiration, and delay browning, softening, and microbial infection [4,5]. Natural biomacromolecule-based edible coatings, particularly proteins, have grown in popularity among researchers due to their film-forming abilities and nutritional value [6]. Whey protein, a by-product of the cheese-making process, has good film-forming and oxygen-inhibiting properties; hence, it is frequently employed to prevent oxidation reactions. Apple, potato, and strawberry respiration and oxidative browning have been greatly reduced by whey protein coatings [7,8]. Additionally, whey protein has been utilized to preserve dried peanuts [9], cake [10], Atlantic salmon [11], meat products [12], and foods high in unsaturated fatty acids [13]. However, due to its high hydrophilicity, the use of whey protein on the surface of fresh-cut fruits and vegetables is limited. As a result, it is critical to use specific modification techniques to improve the functionality of the whey protein film, particularly its water resistance.

The rather weak water vapor resistance of whey protein films can be improved by enzymatic modification. Transglutaminase (TGase), a cross-linked enzyme, can actively promote protein cross-linking to strengthen the protein network, thereby improving the hydrolytic property and water resistance of protein films [14,15]. The efficacy of a protein film’s water resistance has also been improved by the incorporation of numerous hydrophobic compounds, particularly lipids. Although many researchers have investigated the addition of waxes, fatty acids, or acetylated monoglycerides to whey protein films to increase their water resistance [16], the addition of edible vegetable oils to the coating solution is more advantageous in the application scenarios of fresh-cut fruit and vegetable preservation. Sunflower oil is a type of edible oil that is high in linoleic acid and other unsaturated fatty acids. Carotene and vitamin E are also abundant [3]. An innovative preservation and nutritional application system might be created by incorporating TGase and sunflower oil into whey protein film as a layer that can be eaten on the surface of fresh-cut fruits and vegetables.

The purpose of this study was to construct an edible whey protein-based film with good water resistance. The mechanism of whey protein-based complex film formation was studied using particle size distribution, zeta potential, intermolecular force, SDS-PAGE, confocal laser scanning microscopy, and scanning electron microscopy. Furthermore, a suitable whey protein-based emulsion coating was applied in the storage of fresh-cut apples. The preservation effect of the coating treatment was evaluated by measuring the respiration rate, weight loss rate, browning index, enzyme activity, and H_2_O_2_ and MDA contents in apples.

## 2. Materials and Methods

### 2.1. Material

The whey protein isolate (WPI, 95%) was purchased from Fonterra Co-operative Group (Auckland, New Zealand). Sunflower oil was of commercial grade and was produced by Yihai Jiali Golden Arowana Grain, Oil and Food Co., Ltd. (Shanghai, China). Microbial TGase was obtained from Jiangsu Yiming Fine Chemical Industry Co., Ltd. (Qinxing, Jiangsu, China), with an actual activity of 100 U g^−1^. Glycerol (Gly) was used as a plasticizer of analytical grade and was supplied by Tianjin Tianli Chemical Reagent Co., Ltd. (Tianjin, China).

### 2.2. Preparation of Whey Protein-Based Films

The preparation of the film-forming solution was based on Jiang’s method and was appropriately modified [17]. First, a film solution consisting solely of WPI (W) was prepared. WPI was dissolved in distilled water at a concentration of 10% (*w*/*v*) for 30 min with constant magnetic stirring at 250 rpm. The solution was then heated in a water bath at 90 °C for 30 min while continuously stirring and rapidly cooling. Following that, glycerol (60% *w*/*w*) was added to the WPI solution as a plasticizer and stirred for 30 min.

To prepare WPI with TGase film solution (W+TG), TGase at a concentration of 60 U g^−1^ was added to the WPI film solution (pH 7.5) at 40 °C. After 70 min of the enzymatic reaction, it was heated to 90 °C to inactivate the enzyme for 5 min. To prepare WP with TGase and sunflower oil film solutions (WPI+TG+1.5% O/3.0% O), various amounts of sunflower oil (1.5 or 3.0% *w*/*w* based on WPI) were added to the WPI film solution and stirred for 15 min before adding the TGase. Finally, all the film solutions were homogenized for 2 min at 13,500 rpm with Ultra Turrax (IKA Yellowline DI25 basic, IKA, Staufen, Germany).

All the degassed film solutions (10 mL) were poured into a 9 cm internal diameter petri dish and dried at 60 °C for 4 h. Before testing, dried films were peeled and conditioned for 24 h at 25 °C and 50% RH.

### 2.3. Characterization of Film-Forming Solutions

#### 2.3.1. Particle Size Distribution and Zeta Potential

The particle size distribution and the zeta potential of the particles in film-forming emulsions were determined using laser light scattering granulometry [18] on a Malvern Mastersizer Hydro 2000 SM instrument (BT-9300ST), a commercial dynamic light scattering and micro-electrophoresis device (Malvern Zeta mNano ZS, Malvern Instruments, Worcestershire, UK). Emulsion samples were diluted in de-ionized water to 10% of the original concentration and analyzed at 25 °C.

#### 2.3.2. Sodium Dodecyl Sulfate–Polyacrylamide Gel Electrophoresis (SDS-PAGE)

SDS-PAGE was used to evaluate the film-forming solutions under reducing conditions, as described by Zhao [19], with some modifications. A discontinuous system was used, consisting of a 4% (*w*/*v*) acrylamide stacking gel and a 12% (*w*/*v*) acrylamide separating gel. Furthermore, the TGase treatment combined with sunflower oil film-forming solution was centrifuged for 20 min (25 °C, 5000 rpm) to separate the water phase and oil body emulsion. The lower aqueous phase was analyzed directly through SDS-PAGE. The proteins in the upper oil body emulsion then had to be extracted for electrophoretic analysis. The extraction procedure was as follows: SDS was added to the oil body emulsion, which was shaken for 2 min before being centrifuged for 15 min at 4 °C and 14,000 rpm.

### 2.4. Characterization of Films

#### 2.4.1. Film Thickness

The thicknesses of the films were measured using an electronic digital micrometer with a resolution of 0.001 mm [20]. Measurements were taken at 5 points on three films chosen at random for each WPI-based film.

#### 2.4.2. Water Vapor Permeability (WVP)

A modified test based on Butler [21] was used to assess the water vapor permeability of films. Rubber bands around the rim of the cups were used to seal the films over the cups (d = 45 cm). Anhydrous calcium chloride (0% RH, 20 ± 0.1 g) was placed in the cup. The cups were then placed in a distilled water-filled environmental chamber (100% RH, 23 °C). A certain difference in vapor pressure between the inside and outside of the cups was maintained. Anhydrous calcium chloride absorbed water vapor through the test films. Over a 12 h period, cups were weighed every 3 h. WVP (g mm m^−2^ h^−1^ kPa^−1^) was calculated using the following formula:(1)$$WVP=\frac{\Delta m\cdot d}{A\cdot t\cdot \Delta P}$$
where Δ*m* is the weight of water vapor permeated (g), *d* is the film thickness (mm), *A* is the area of exposed film (mm^2^), *t* is the time, and ΔP is the water vapor pressure difference across the film.

#### 2.4.3. Moisture Content (MC), Moisture Absorption (MA), and Water Solubility (WS)

The moisture content, moisture absorption, and water solubility of films were determined using Oliveira’s method [22]. The moisture content of the composite film was obtained by weighing it before and after drying (at 105 ± 1 °C until a constant weight). The weight gain variations of films transferred from 0% RH to 55% RH were used to determine the moisture absorption of the composite films. Water solubility was calculated as a percentage of the film’s soluble dry matter content after 24 h in water.

#### 2.4.4. Tensile Strength (TS) and Elongation at Break (EB)

Tensile strength and elongation at break of the films (10 × 50 mm) were measured using an ASTM standard method D882 on a Texture Analyzer TA-XT2i (Stable Microsystems, Haslemere, UK), and each sample was paralleled 5 times. The initial separation distance was 30 mm, and it stretched at a rate of 1.0 mm s^−1^ until it broke. The TS and EB were calculated according to the following equations:(2)$$TS=\frac{F}{L\cdot X}$$
where *F* is the maximum force at rupture of the film (N), *L* is the film width (mm), and *X* is the film thickness (mm).
(3)$$EB=\frac{L_{1}-L_{0}}{L_{0}}\times 100\%$$
where *L*_1_ is the film elongation at rupture (mm) and *L*_0_ is the film’s initial gage length (mm).

### 2.5. Intermolecular Forces of WPI-Based Films

Protein interactions in whey protein films were assessed by using the method described by Chawla [23], which assumes that proteins have varying solubilities in different solvent systems. Tris-HCl buffer, urea, sodium chloride solution, SDS, and β-mercaptoethanol were chosen as reagents. Protein was extracted in its natural state using Tris-HCl buffer, urea and SDS were able to destroy non-covalent interactions, and β-mercaptoethanol was a reducing agent capable of destroying disulfide bonds. Changes in protein interactions are assessed by adding or subtracting several reagents. The reagents are as follows: S_1_: 0.6 mol L^−1^ NaCl; S_2_: 20 mmol L^−1^ Tris-HCl (pH 8.0); S_3_: 20 mmol L^−1^ Tris-HCl containing 1% SDS (*w*/*v*) (pH 8.0); S_4_: 20 mmol L^−1^ Tris containing 1% SDS and 8 mol L^−1^ urea (pH 8.0); S_5_: 20 mmol L^−1^ Tris containing 1% SDS, 8 mol L^−1^ urea, and 2% β-mercaptoethanol (*v*/*v*) (pH 8.0).

An amount of 0.2 g protein film sample was mixed with 4 mL of the above 5 reagents and shaken for 1 min before being bathed in water at 40 °C for 4 h. After 30 min of centrifugation at 12,000× *g*, 0.5 mL supernatant was added to 0.5 mL of 50% (*w*/*v*) TCA and stored at 4 °C for 18 h. The supernatant was poured after centrifugation at room temperature for 15 min at 10,000× *g*, and 1 mL of 10% (*w*/*v*) TCA was added for washing once. After 15 min of centrifugation at 15,000× *g*, the supernatant was poured, and 3.5 mL of 0.5 mol L^−1^ NaOH was precipitated to dissolve. The protein content was determined by the biuret method. In addition, as a reference amount of protein, 0.2 g of each sample was added to 4 mL of 0.5 mol L^−1^ NaOH.

### 2.6. Microstructure of WPI-Based Films

#### 2.6.1. Confocal Laser Scanning Microscopy (CLSM)

The micro-morphology of emulsion refers to the method of Ma [24]. The film-forming solution was stained with Nile blue and Nile red; then, the stained emulsion sample was placed on the slide and examined with an argon/krypton laser, with an excitation line of 514 nm and a helium–neon laser (HeNe) with excitation at 633 nm.

#### 2.6.2. Scanning Electron Microscopy (SEM)

According to the method of Galus and Kadzinska [25], the surface and cross-section morphology of the gold-sprayed film sample were analyzed using a field emission scanning electron microscope (JSM-7800F, Japanese Electronics Co., Ltd., Tokyo, Japan). The film sample was placed on the sample table with conductive silica gel and observed at a magnification of ×1000 (surface) or 1500 (cross-section).

### 2.7. Fresh-Cut Apple Coating Treatment and Storage

#### 2.7.1. Fruit Treatment

Apples (*Yantai Fuji*) in the commercial ripening stage were obtained from a local market. Apples of comparable size and color were chosen and chilled for 24 h in a refrigerator at 4 °C. The surface of the apples was sterilized with a sodium hypochlorite solution (200 μL L^−1^) for 2 min before being washed with distilled water. The apples were then peeled with a pre-sterilized knife, the seeds were removed, and they were divided evenly into 12 portions. The apples were randomly divided into two groups: one was soaked in distilled water for 2 min (CK), and the other was soaked in the coating solution (WPI+TG+1.5% O) for 2 min (CF). Following the drying of the surface coating solution, the two groups of apples were stored at 8 °C and tested every two days.

#### 2.7.2. Weight Loss (WL), Respiration Intensity, and Browning Index (BI)

Measurement of Weight Loss Rate

The initial mass of the apples was recorded as *M*_0_ on the first day of the experiment, and the mass of the fresh-cut apples on the test day after a certain storage period was recorded as *M*_1_. Then the weight loss rate can be expressed as:(4)$$WL=\frac{M_{0}-M_{1}}{M_{0}}\times 100\%$$

Determination of Respiratory Intensity

Apples (100 g) were placed in different breathing chambers, and the amount of CO_2_ mg released per kg of fresh-cut apples per hour was measured and calculated to determine their respiration intensity.
(5)$$RI=\frac{\left(V_{2}-V_{1}\right)\times M\times 44}{m\times t}$$
where *V*_1_ (mL) is the volume of oxalic acid consumed when titrating the sample; *V*_2_ (mL) is the volume of oxalic acid consumed when titrating blank; *M* (mol L^−1^) is the molarity of oxalic acid; *m* (Kg) is the ample quality; *t* (h) is the determination of the time.

Determination of Browning Index (BI)

The browning index of fresh-cut apples was calculated using Jiang’s method [26]. A colorimeter (CR-20, Konica Minolta, Tokyo, Japan) was used to measure color parameters. The following is how the BI was calculated:(6)$$BI=\frac{X-0.31}{0.172}\times 100$$
(7)$$X=\frac{a+1.75L}{5.645L+a-3.02b}$$

#### 2.7.3. Determination of PPO and MDA

The PPO activity of fresh-cut apples was determined according to the method of Galeazzi [27]. Five grams of the sample was weighed and ground into a pulp before adding 20 mL of phosphoric acid buffer solution, mixing, and centrifuging (4 °C, 12,000× *g*). The PPO activities were determined using the supernatant (raw enzyme extract). A 3 mL phosphate buffer solution (500 mmol L^−1^ catechol) was added to 1 mL crude enzyme extraction and measured at 420 nm. Each sample was taken three times in parallel. The amount of enzyme required to cause a change in absorbance value of 0.01 per minute is expressed as enzyme activity.

The MDA concentration was determined using the thiobarbituric acid (TBA) method described by Wang [28]. One gram of pulp sample was weighed and extracted with 5% (*m*/*v*) trichloroacetic acid solution before centrifuging at 4 °C at 10,000 rpm for 15 min. The supernatant was then mixed with 2 mL of 0.6% thiobarbituric acid solution and bathed at 100 °C for 30 min. After cooling, the absorbance values at 450 nm, 532 nm, and 600 nm were measured.
(8)$$MDA=6.45\times \left(A_{532}-A_{600}\right)-0.56\times A_{450}$$

#### 2.7.4. Determination of H_2_O_2_ and CAT

The H_2_O_2_ concentration and CAT activity were determined using the method described by Zhao [29]. Five grams of each sample were ground into a pulp before being added to the extract and centrifuged (4 °C, 12,000× *g*). The supernatant was added to the reaction solution, and the absorbance at the corresponding wavelength was measured.

### 2.8. Statistical Analysis

Three independent experiments were carried out, with samples collected at least in triplicate samples in each run. SPSS Statistics 20 was used for statistical analysis, and GraphPad Prism 5 was used for graphing. Duncan’s multiple range tests and one-way analysis of variance (ANOVA) (*p* < 0.05) were used to determine the differences in the results among the different samples.

## 3. Results and Discussion

### 3.1. The Effects of TGase and Sunflower Oil Content on the Physical Properties of WPI-Based Films

The effects of TGase and sunflower oil content on the water resistance and mechanical properties of WPI-based films are shown in Table 1. The WVP of the WPI film modified with TGase decreased significantly, and it decreased even more after the addition of sunflower oil. A small amount of sunflower oil (1.5%) can be added to the film matrix to improve the composite film’s water vapor barrier capacity and reduce water diffusion by combining the hydrophobic phase. This is consistent with the findings of Valenzuela [30]. When sunflower oil was added to quinoa protein, its WVP decreased significantly. Other lipids, such as olive oil [31], rosemary essential oil [32], and oregano essential oil [22], had the same effect when added to the whey protein film. However, when 3% sunflower oil was added instead of the 1.5% supplemental level of sunflower oil, the WVP rate increased. The WVP of the edible film is heavily influenced by particle size distribution, which is affected by homogenization methods and conditions. The smaller the particle size and the more uniform the distribution, the better the film’s water-blocking performance [33,34]. When too many lipids are added, a portion of the oil accumulates, affecting the uniformity of the grease distribution on the film and decreasing water resistance.

The amount of water molecules that occupy the hollow positions in the bio-composite film network structure determines the moisture content of the film, which affects the macroscopic performance of the film [35]. The water content of the whey protein composite film was significantly reduced when sunflower oil was added (Table 1). This is primarily because oil not only takes up the original position of water molecules in the protein network structure, reducing the available space for water molecules, but parts of the protein–water interaction also become replaced by protein–oil interactions, lowering the moisture content of the film.

The hydrophilicity of the whey protein, the H-group in the water molecule, and the OH-group in the glycerol in the biopolymer chain are usually attributed to the water solubility and absorption of the whey protein film [13,32]. After the TGase modification treatment, the water solubility and absorption of the WPI-based film gradually decreased. With the addition of sunflower oil, the water solubility of the film decreased gradually as the amount added increased (Table 1).

The mechanical strength of the film includes TS and EB. TGase could catalyze the binding reaction between the ε-amino group in lysine and the γ-hydroxyamide group in a glutamic acid to form intramolecular or intermolecular ε-(γ-glutamine) amide lysine covalent bonds, resulting in intramolecular or intermolecular cross-linking of proteins. Protein cross-linking led to enhanced mechanical properties of whey protein membranes [36]. After adding 1.5% sunflower oil, the TS of the whey protein composite film increased significantly. However, as the amount of sunflower oil increased, both the EB and the TS decreased significantly (Table 1). This may be because of the composition of emulsion films. Lipid molecules filled the protein matrix in these structures. The interactions between polar and lipid molecules in those structures appeared to be weaker than those between simply polar molecules in control films [37].

### 3.2. Particle Size Distribution and Zeta Potential of WPI-Based Film-Forming Solutions

Many physical and chemical properties of the edible film are affected by the particle size and disruption of the emulsion. As shown in Figure 1A,B, the single whey protein has a small particle size (3.2 μm) and a unimodal distribution; after TGase cross-linking, the average particle size increases significantly to 6.2 μm and exhibits a bimodal distribution, indicating that TGase has a cross-linking action on whey protein. Cross-linking causes the production of β-lactoglobulin dimers, trimers, and even multimers, resulting in the formation of large-molecule proteins [38,39], as well as an increase in particle size and another peak. When compared to the TGase cross-linking without oil, the average particle size of the whey protein emulsion with 1.5% sunflower oil was significantly reduced to 4.2 μm. Because some proteins interact with the oil, the cross-linking of proteins is hampered, causing the peak to shift to the left and become more concentrated. The oils then accumulate and increase the particle size as the amount of sunflower oil added increases. According to Galus [37], regardless of whether almond or walnut oil is added to whey protein, increasing the amount of oil added increases the average particle size of the coating liquid.

Different interactions between the oil and protein may occur in order to form an effective edible film and coating. This interaction can be understood by examining the zeta potential of the emulsion. Figure 1C depicts the effect of various treatment methods on the potential of the film-forming fluid. Except for the positive value of pure whey protein, the other values are negative. This is because 60% of the whey protein is β-lactoglobulin (β-Lg) and 20% is α-lactalbumin (α-La), which have isoelectric points of 5.2 and 4.1, respectively, and a solution environment of pH 7.0. When whey protein molecules are cross-linked by the addition of TGase, the protein structure changes, and the potential becomes negative. When sunflower oil is added, a portion of the protein molecules rise to the oil–phase interface, its structure expands to some extent, and molecular rearrangement occurs, exposing acidic amino acids; these acidic amino acids are negative, so the negative potential increases [40].

### 3.3. SDS-PAGE Analysis of WPI-Based Film-Forming Solutions

Whey protein contains a high concentration of free sulfhydryl groups and disulfide bonds. The degree of cross-linking of TGase to whey protein, as well as the distribution of protein components in the emulsion system, can be determined using polyacrylamide gel electrophoresis. In Figure 2A, it is shown that, compared with band 2, the number of α-lactalbumin dimers and β-lactoglobulin dimers increased significantly in band 3, and large molecular weight polymers also appeared, confirming the cross-linking effect of TGase on whey protein. Bands 4 and 5 showed no significant difference when compared to band 3. Therefore, the proteins in the emulsion’s water phase and oil body emulsion were separated for further study. Bands 4 and 7 in Figure 2B represent the electrophoresis spectrum of the oil body emulsion. The figure shows that a portion of β-lactoglobulin, α-lactalbumin dimer, and β-lactoglobulin exist primarily in the oil body emulsion, indicating that proteins with molecular weights ranging from 18 to 36 kDa tend to become interface proteins and stabilize the emulsion droplets.

### 3.4. Intermolecular Force Analysis of WPI-Based Films

Hydrogen bonds, disulfide bonds, electrostatic interactions, hydrophobic interactions, and ionic bonds are the most common interactions between protein molecules. Their formation and distribution can keep the protein’s three-dimensional network structure intact during the film’s formation process [20]. The solubility of these proteins in different solvent systems can express the type and size of their intramolecular or intermolecular interaction forces. Urea tends to break hydrogen bonds, SDS tends to break hydrophobic bonds, and β-mercaptoethanol can reduce disulfide bonds. Table 2 shows that the protein solubility of all TGase-modified films in S_5_ increased when compared to the single WPI film, indicating increased disulfide bonds in the films. However, after the addition of 3% sunflower oil, the solubility of the composite film in S_5_ decreased, as confirmed by the electrophoresis results. To some extent, the addition of oil inhibits the cross-linking effect of TGase.

### 3.5. The Effects of TGase and Sunflower Oil Content on the Microstructures of WPI-Based Films

We can clearly see from Figure 3A(a,b) that when sunflower oil is not added, there is only a continuous protein phase. After the addition of sunflower oil, the oil interacts with the protein to form emulsion droplets (Figure 3A(c,d)). The size and quantity of emulsion droplets gradually increase as the oil content increases (Figure 3A(e,f)), which is consistent with the particle size analysis results.

The untreated whey protein film has a smooth surface with no wrinkles or cracks, and the inside is flat and free of bubbles (Figure 3B(a,b)). After being modified with TGase, the structure of the film becomes denser with obvious protrusions, and the protein network structure becomes more compact (Figure 3B(c,d)). When 1.5% sunflower oil is added to the film, the surface remains smooth, but the inside has wrinkles, and the emulsion droplets are evenly distributed in the protein network (Figure 3B(e,f)). However, as the oil content increased to 3%, wrinkles appeared on the film surface and more oil droplet aggregates were formed, which were irregularly separated in the whey protein film (Figure 3B(g,h)). These results suggest that the interaction of oil and protein is insufficient to stabilize the oil droplets in the emulsion. Other studies, such as those on olive oil [31] and rapeseed oil [41], have found a similar situation.

### 3.6. Hypothetical Schematic Images for the Formation Mechanisms of Whey Protein-Based Complex Film

Natural globular whey proteins are properly heated to expose the hydrophobic amino acids within their structures, allowing them to form films. Some new disulfide and hydrogen bonds were generated by TGase modification, resulting in a more compact three-dimensional protein network structure, which improves the performance of the whey protein-based complex film (Figure 4B). When sunflower oil is added, some take up the original positions of the water molecules in the protein network, while others interact with the proteins to form emulsion droplets. β-lactoglobulin, α-lactalbumin dimer, and β-lactoglobulin dimer with molecular weights of 18–36 kDa tend to act as interfacial proteins. The sunflower oil embedded in the protein network structure, as well as the dispersed emulsion droplets, improve the water resistance of the whey protein-based complex film (Figure 4C). As a result, this WPI+TG+1.5% O film can be used in high-moisture preservation situations and demonstrates a good preservation effect. However, when the amount of sunflower oil added was increased, the oil droplets themselves aggregated. The TGase cross-linking effect on the protein was inhibited, which had a negative impact on the performance of the whey protein-based film (Figure 4D).

### 3.7. WPI-Based Emulsion Coating in Fresh-Cut Apple Preservation

#### 3.7.1. Browning Index (BI), PPO Activity, and MDA Content

Enzymatic browning is the main cause of the browning of fresh-cut fruits and vegetables. The fresh-cut apples showed browning during storage regardless of whether they were uncoated or coated, as shown in Figure 5A,B. However, a WPI-based emulsion coating treatment significantly reduced the browning of fresh-cut apples. The browning index of fresh-cut apples after the coating treatment was reduced by 46.39% when compared to the control group. This is related to the whey protein film’s barrier properties. On the other hand, it is possible that the cysteine side chain residues in whey protein can inhibit the polyphenol oxidase-mediated enzymatic reaction, reducing browning [34].

PPO is an enzyme that can hasten the onset of enzymatic browning. It can catalyze the o-hydroxylation of monophenols to o-diphenol, which can then be oxidized to dark quinone [42]. Enzymatic browning is closely related to PPO activity in fresh-cut fruits and vegetables during the post-harvest processing and storage process. As shown in Figure 5C, the emulsion coating treatment significantly reduced the increase in PPO activity over the entire storage period.

Lipid peroxidation is an important indicator of a deteriorated membrane system and cell metabolism. MDA is produced in the plant cell membrane as a result of lipid peroxidation. MDA buildup destroys the composition of the cell membrane, promoting the accumulation of brown polymers and causing browning and quality decline in fruits. As a result, MDA content is an important index for determining the degree of lipid peroxidation in the membrane of fresh-cut apples [1]. The effects of the emulsion coating treatment on the content of MDA in fresh-cut apples are shown in Figure 5D. The MDA content in both the coated and uncoated groups increased and then decreased, reaching a maximum on the sixth day of 3.76 μmol/kg and 2.98 μmol/kg, respectively. The MDA content in coated fresh-cut apples decreased by 21.74% when compared to the uncoated group. The content of MDA was significantly reduced by the coating treatment at the early stage of storage (0–4d), but the difference was extremely significant at the late stage (6–8d). These findings indicate that the coating treatment could significantly reduce the MDA content increase, protect the cell membrane of fresh-cut apples from damage, and reduce browning.

#### 3.7.2. Weight Loss (WL) and Respiration Intensity

Due to mechanical damage and transpiration during processing, water and weight are easily lost during the storage of fresh-cut fruits and vegetables [42]. Although the weight loss in each group increased significantly with storage time, at the end of storage the weight loss in the coated group was significantly (26.55%) lower than that in the uncoated group (Figure 6A). It can be concluded that the modification of TGase and the addition of 1.5% sunflower oil could improve the water resistance performance of the WPI-based coating so that it can effectively prevent the water loss of fresh-cut apples. When the coating material contains lipids, the coating treatment can prevent water loss, and the higher the lipid content, the more noticeable the effect [34].

Due to peeling and cutting operations, the plant tissues become subjected to physiological pressures, which increase the rate of respiration and metabolism for processed fresh-cut apples, resulting in “breath injury”. The respiration intensity of uncoated fresh-cut apples showed a slight upward trend. On the sixth day, there was an obvious respiratory peak of 114 CO_2_ mg kg^−1^ h^−1^, and the respiration intensity showed a downward trend on the last day. Similar findings were reported in the study of Salvia [43]. During the whole storage period, the respiration intensity of fresh-cut apples was significantly inhibited by the WPI-based coating (Figure 6B).

#### 3.7.3. H_2_O_2_ Content and CAT Activity

H_2_O_2_ accumulated during the storage of fruits and vegetables can directly or indirectly lead to lipid peroxidation damage of the cell membrane, increase cell membrane permeability, and thus accelerate cell aging and disintegration [29]. As shown in Figure 7A, in both coated and uncoated groups, the H_2_O_2_ content for all apples presents a rising trend during the whole storage period. However, in the late storage period (6 d), the uncoated group’s H_2_O_2_ content reached 78 µmol/g, and the coating treatment significantly inhibited the rising trend, reducing H_2_O_2_ content by 25.32%. This may be due to the barrier effect of the WPI-based coating, which reduces the respiration rate of the fruit, inhibiting the oxidative stress response of fresh-cut apples.

Catalase is an iron-containing hemoglobin protease found in almost all living organisms that rely on oxygen. It can catalyze the breakdown of accumulated hydrogen peroxide in plants into water and oxygen, reducing the oxidative damage that H_2_O_2_ can cause to fruit and vegetable tissues [44]. As shown in Figure 7B, the increasing trend of CAT activity in the uncoated group was roughly consistent with that of the H_2_O_2_ content and peaked on the sixth day. After coating the fresh-cut apple, the activity of CAT was significantly reduced. The lower CAT activity could be attributed to the lower H_2_O_2_ content of fresh-cut apples due to the WPI-based coating which has an oxygen barrier [45].

## 4. Conclusions

In this study, an edible whey protein-based film with good water resistance was produced. Meanwhile, the film-forming mechanisms and preservation effect of the whey protein-based film on fresh-cut apples were verified. Sunflower oil (add 1.5% *w*/*w*) could interact with β-lactoglobulin, α-lactoglobulin dimer, and β-lactoglobulin dimer to form emulsion droplets that are evenly dispersed throughout the protein film. This effect, combined with the covalent cross-linking of TGase, significantly improves the films’ mechanical properties and water resistance. However, too much and unevenly distributed sunflower oil (add 3% *w*/*w*) partially prevented the covalent cross-linking of TGase, reducing the elongation at the break of the composite film. The best composite coating treatment significantly reduced the weight loss rate and browning index of coated fresh-cut apples by 26.55% and 46.39%, respectively. This was accomplished by the coating treatment significantly inhibiting the respiration rate increase, PPO and CAT activity enhancement, H_2_O_2_ production, and MDA accumulation.

## Figures and Tables

**Figure 1 foods-12-01140-f001:**
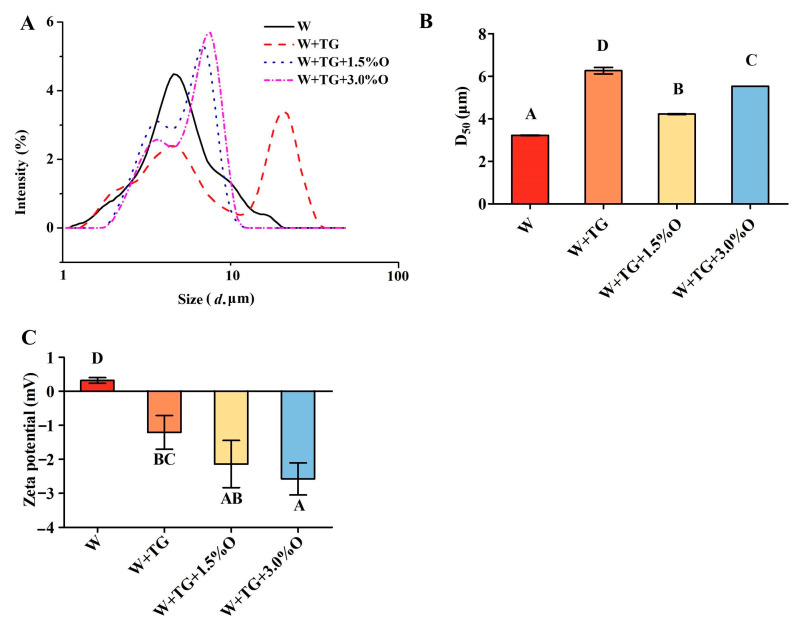
Particle size and zeta potential analyses of WPI-based film-forming solutions. (**A**) Particle size distribution; (**B**) mean particle diameter; (**C**) zeta potential.W means the film solution consisting solely of WPI; W+TG denotes WPI with MTGase; WPI+TG+1.5% O denotes WPI with MTGase and 1.5% sunflower oil; WPI+TG+3.0% O denotes WPI with MTGase and 3.0% sunflower oil. Data are presented as means ± standard deviation. The different uppercase letters indicate significant differences within the different groups (*p* < 0.05).

**Figure 2 foods-12-01140-f002:**
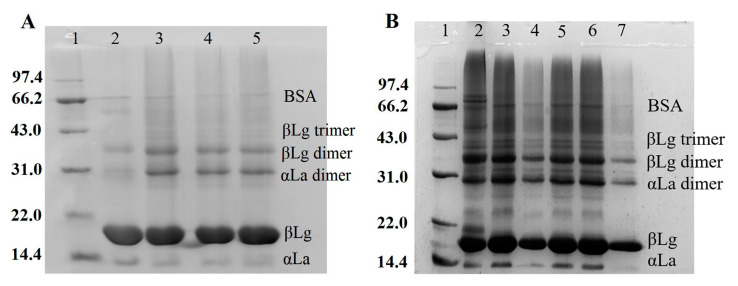
SDS-PAGE profiles of the proteins in WPI-based film-forming solutions. (**A**) SDS-PAGE analysis of the proteins in film-forming solutions (1: marker, 2: W, 3: W+TG, 4: W+TG+1.5% O, 5: W+TG+3.0% O); (**B**) SDS-PAGE analysis of the proteins in water phase and oil body emulsion (1: marker, 2: W+TG+1.5% O, 3: the water phase of W+TG+1.5% O, 4: the oil body emulsion of W+TG+1.5% O, 5: W+TG+3.0% O, 6: the water phase of W+TG+3.0% O, 7: the oil body emulsion of W+TG+3.0% O).

**Figure 3 foods-12-01140-f003:**
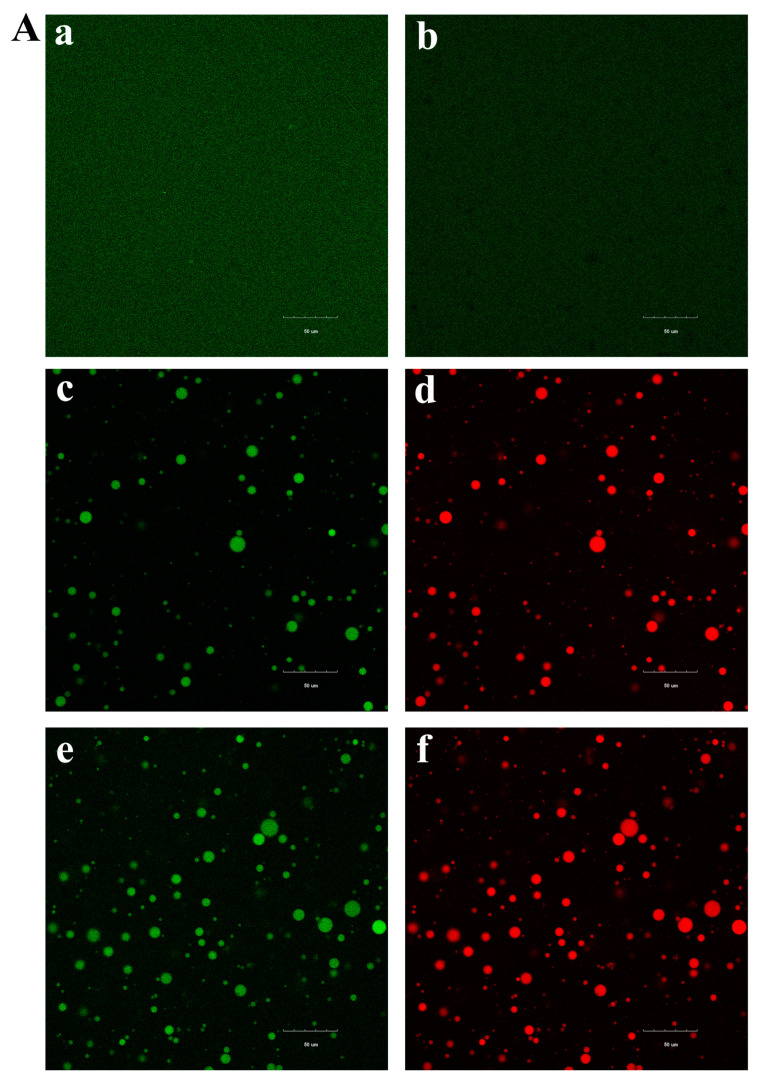
Microstructure of WPI-based film-forming solutions and WPI-based films. (**A**) Confocal laser scanning microscopy (**a**: W; **b**: W+TG; **c**,**d**: W+TG+1.5% O; **e**,**f**: W+TG+3.0% O); (**B**) scanning electron micrographs of surfaces (**a**,**b**: W; **c**,**d**: W+TG; **e**,**f**: W+TG+1.5% O; **g**,**h**: W+TG+3.0% O) and cross-sections (**a**,**b**: W; **c**,**d**: W+TG; **e**,**f**: W+TG+1.5% O; **g**,**h**: W+TG+3.0% O). W means the film solution consisting solely of WPI; W+TG denotes WPI with MTGase; WPI+TG+1.5% O denotes WPI with MTGase and 1.5% sunflower oil; WPI+TG+3.0% O denotes WPI with MTGase and 3.0% sunflower oil. The graphic scale of CLSM photos is 50 μm. SEM surface photographs have a 1000× magnification, whereas cross-section photos have a 1500× magnification.

**Figure 4 foods-12-01140-f004:**
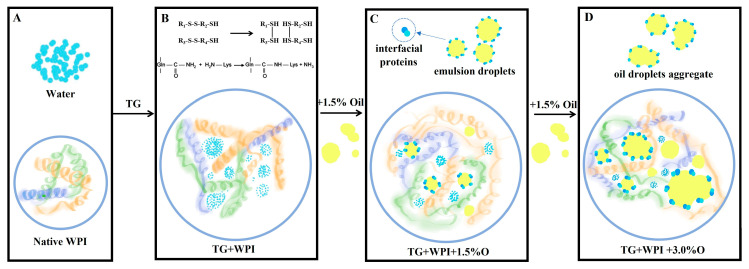
Hypothetical schematic images for the formation mechanisms of whey protein-based complex film. (**A**) Native WPI; (**B**) WPI with MTGase; (**C**) WPI with MTGase and 1.5% sunflower oil; (**D**) WPI with MTGase and 3.0% sunflower oil.

**Figure 5 foods-12-01140-f005:**
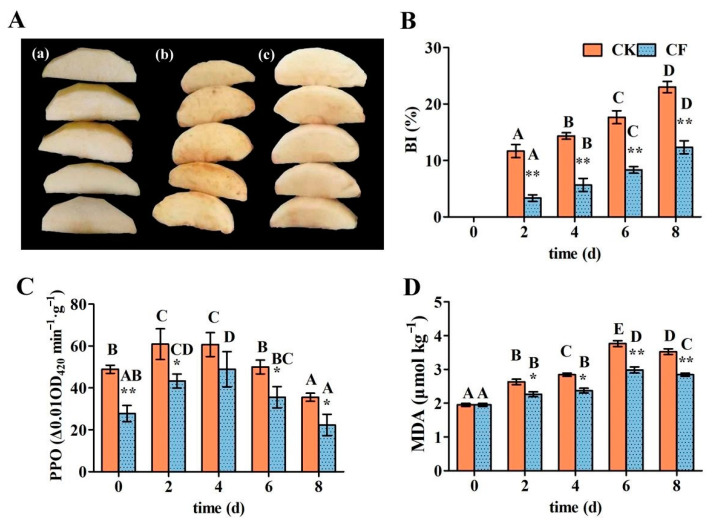
The effects of emulsion coating treatment on the physicochemical quality of fresh-cut apples. (**A**) Appearance of fresh-cut apples ((**a**) fresh-cut apples; (**b**) CK at 8 d; (**c**) CF at 8 d); (**B**) browning index; (**C**) PPO activity; (**D**) MDA content. *Note*: CK means uncoated fresh-cut apples; CF means coated fresh-cut apples. Data are presented as means ± standard deviation. The different uppercase letters in the same group indicate significant differences within the different storage times (*p* < 0.05). * in the same storage time indicates significant differences within the different groups (*p* < 0.05). ** in the same storage time indicates significant differences within the different groups (*p* < 0.01).

**Figure 6 foods-12-01140-f006:**
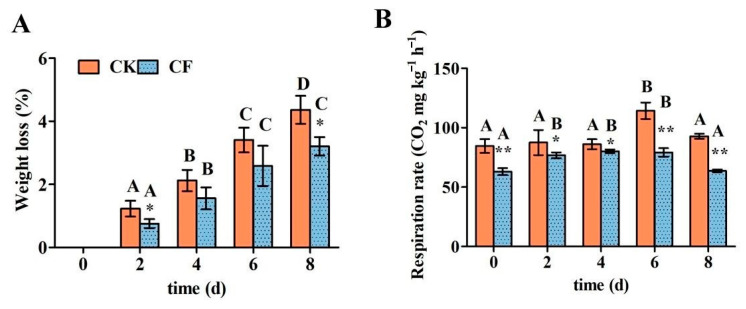
The effects of emulsion coating treatment on the weight loss and respiration intensity of fresh-cut apples. (**A**) Weight loss; (**B**) respiration intensity. *Note*: CK means uncoated fresh-cut apples; CF means coated fresh-cut apples. Data are presented as means ± standard deviation. The different uppercase letters in the same group indicate significant differences within the different storage times (*p* < 0.05). * in the same storage time indicates significant differences within the different groups (*p* < 0.05). ** in the same storage time indicates significant differences within the different groups (*p* < 0.01).

**Figure 7 foods-12-01140-f007:**
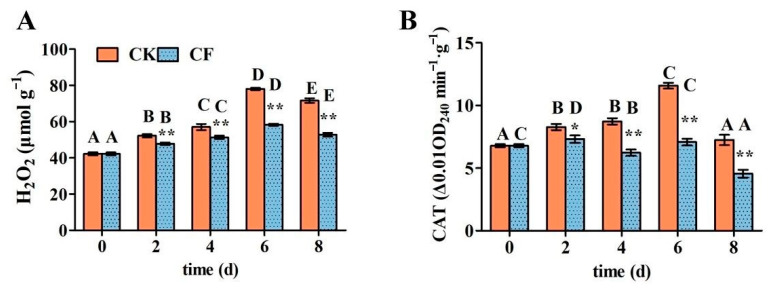
The effects of emulsion coating treatment on the H_2_O_2_ content and CAT activity of fresh-cut apples. (**A**) H_2_O_2_ content; (**B**) CAT activity. *Note*: CK means uncoated fresh-cut apples; CF means coated fresh-cut apples. Data are presented as means ± standard deviation. The different uppercase letters in the same group indicate significant differences within the different storage times (*p* < 0.05). * in the same storage time indicates significant differences within the different groups (*p* < 0.05). ** in the same storage time indicates significant differences within the different groups (*p* < 0.01).

**Table 1 foods-12-01140-t001:** The effects of TGase and sunflower oil content on the physical properties of WPI-based films.

Samples	W *	W+TG	WPI+TG+1.5% O	WPI+TG+3.0% O
	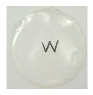	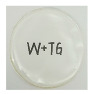	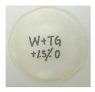	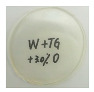
Thickness (μm)	160.31 + 7.52 ^a^	163.60 + 9.08 ^a^	183.17 + 4.08 ^b^	205.04 + 6.41 ^c^
WVP (g·mm·m^−2^·h^−1^·kPa^−1^)	1.38 + 0.161 ^c^	1.21 + 0.05 ^b^	0.48 + 0.09 ^a^	0.59 + 0.01 ^b^
Moisture content (%)	26.79 + 0.68 ^c^	23.72 + 1.15 ^b^	21.88 + 0.68 ^a^	21.02 + 0.90 ^a^
Water solubility (%)	65.30 + 2.07 ^d^	55.57 + 1.66 ^c^	49.65 + 1.09 ^b^	43.11 + 0.88 ^a^
Moisture absorption (%)	6.35 + 0.17 ^c^	4.87 + 0.20 ^b^	3.95 + 0.28 ^a^	3.70 + 0.36 ^a^
Elongation at break (%)	37.94 + 3.75 ^b^	48.25 + 5.17 ^c^	44.43 + 6.73 ^bc^	29.09 + 5.74 ^a^
Tensile strength (MPa)	2.58 + 0.20 ^a^	2.99 + 0.14 ^ab^	3.13 + 0.24 ^b^	3.04 + 0.29 ^ab^

Data are presented as means ± standard deviation. Different superscript letters within the same row indicate significant differences between the films (*p* < 0.05). * W: WPI film; W+TG: WPI with MTGase film; WPI+TG+1.5% O: WPI with MTGase and 1.5% sunflower oil film; WPI+TG+3.0% O: WPI with MTGase and 3.0% sunflower oil film.

**Table 2 foods-12-01140-t002:** The intermolecular forces of WPI-based films.

Samples	W *	W+TG	W+TG+1.5%O	W+TG+3.0%O
S1	38.78 ± 0.78 ^c^	35.34 ± 5.29 ^b^	35.90 ± 5.29 ^b^	31.73 ± 1.55 ^a^
S2	45.67 ± 4.72 ^b^	42.34 ± 6.91 ^ab^	36.35 ± 1.17 ^a^	35.75 ± 2.35 ^a^
S3	50.07 ± 4.14 ^c^	43.75 ± 1.52 ^ab^	39.79 ± 0.78 ^a^	46.43 ± 5.57 ^bc^
S4	59.75 ± 1.26 ^a^	63.12 ± 1.80 ^b^	62.95 ± 3.38 ^b^	67.75 ± 6.72 ^c^
S5	61.76 ± 6.77 ^a^	96.09 ± 8.64 ^c^	89.43 ± 7.05 ^bc^	83.28 ± 3.06 ^b^

Data are presented as means ± standard deviation. Different superscript letters within the same row indicate significant differences between the films (*p* < 0.05). * W: WPI film; W+TG: WPI with MTGase film; WPI+TG+1.5% O: WPI with MTGase and 1.5% sunflower oil film; WPI+TG+3.0% O: WPI with MTGase and 3.0% sunflower oil film.

## Data Availability

All raw data used for figure and table generation in this manuscript can be obtained by contacting the corresponding author.

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
