# Peer review of "Application of Whey Protein-Based Emulsion Coating Treatment in Fresh-Cut Apple Preservation"

_foods, 2023, doi:10.3390/foods12061140_

Round 1

Reviewer 1 Report

The paper ‘Application of whey protein-based emulsion coating treatment in fresh-cut apple preservation’ by Xin et al. is designed to produce an edible whey protein-based film with good water resistance. In this work, the mechanism of whey protein-based complex film formation was studied using particle size distribution, zeta potential, intermolecular force, SDS-PAGE, confocal laser scanning microscope, and scanning electron microscope. Furthermore, a suitable whey protein-based emulsion coating was applied in the storage of fresh-cut apples. The preservation effect of the coating treatment is evaluated by measuring the respiration rate, weight loss rate, browning index, enzyme activity, and H2O2 and MDA contents in apples. I found the work quality excellent, however the following points need to be addressed:

Section 2.2. The preparation of whey protein-based films needs a reference.

Explain which diluent (if there is any) did you use to determine the z-potential and probably particle size of the samples?

Before the SDS-PAGE assay you need to quantify the protein content, have you measured them? If so, how much protein did you load in each well?

The scale bars are not obvious in Figure 3 for both confocal and SEM micrographs, would you mind splitting it into two separate figures to see the details?

Reviewer 2 Report

Ref: MS titled “Application of whey protein-based emulsion coating treatment in fresh-cut apple preservation”. The theme of the work is interesting and is of interest to the readers of the journal. The manuscript contains some new information and interesting data about the possible potential of whey protein-based emulsion coating on fresh-cut apple fruit. The keywords should be different from the words used in title. The introduction is well written and methodology is reproducible. Please add maturity indices of apple fruit at which stage these were harvested. Results are well written but discussion needs improvement. Conclusion should be shortened.  
